# Mitigating Unobserved Confounding via Diffusion Probabilistic Models

## Abstract

Learning Conditional average treatment effect estimation from observational data is a challenging task due to the existence of latent covariates. Previous methods mostly focus on assuming the ignorability assumption ignoring the latent covariates or overlooking the impact of an apriori knowledge on the generation process of the latent variable, which can be quite impractical in real-world scenarios. We introduce a novel framework that mitigates unobserved confounding by generating the latent covariates using a conditional diffusion probabilistic model. This model first infers a causal context vector from the observed data, and then uses this vector to guide a reverse diffusion process that synthesizes the unobserved covariate. We render this architecture tractable by deriving a closed-form variational lower bound for its optimization. To ensure causal validity, we theoretically analyze that the latent variable $z$ learned by our model is orthogonal-identifiable. In the experiments, we compare our model with the state-of-the-art methods based on two standard benchmarks, demonstrating consistent improvements of our model.

## 1 Introduction

Estimating the Conditional Average Treatment Effect (CATE) from observational data is a fundamental problem in a multitude of fields, including personalized medicine, economic policy-making, and online advertising. The ability to accurately predict how an individual will respond to a specific treatment or intervention allows for more effective, data-driven decision-making. However, the validity of such estimations is often compromised by confounding bias, which arises when extraneous variables, known as covariates or confounders, are associated with both the treatment assignment and the outcome. While numerous methods have been developed to adjust for observed covariates, the presence of latent covariates remains a critical and pervasive challenge.

To improve the CATE estimation, a line of methods uses representation learning to force the covariate balance. The representation learning-based approaches aim to generate covariate representations that eliminate the differences within the distributions between treatment and control groups to mitigate confounding bias. To obtain the balanced representations, integral probability metric (IPM) for regularization Johansson et al. (2016), local similarity preservation Yao et al. (2018), targeted learning Zhang et al. (2020a), and optimal transport (Wang et al., 2023) have been adopted. These methods have achieved state-of-the-art performance but operate under the strong, and often untestable, assumption of *ignorability*, which posits that all confounding variables are observed. In many real-world scenarios—such as clinical studies where a patient's genetic predispositions or lifestyle factors are unrecorded—this assumption is violated, leading to biased estimates and potentially flawed conclusions.

To address the issue of unobserved confounding, some methods that only rely on large-scale observation data (OBS) have gathered increasing attention, including sensitivity analysis, instrumental variables, and front-door adjustment methods. However, these methods require strong assumptions. On the other hand, some prominent generative models have been proposed to generate such latent covariates that we could utilize them to isolate the causal effect of treatment on outcome. For instance, VAE-based method CEVAE (Louizos et al., 2017) assumes that there exists a proxy variable in the causal graph, and then generates the unmeasured confounder by optimizing the variational lower bound of this graphical model, GANITE (Yoon et al., 2018) aims to generate the counterfactual distributions using GAN, and accordingly to infer the CATE in an unbiased setting. Other exemplar

methods involve generating the unmeasured confounder with Gaussian Processes (Witty et al., 2020), Imitation Learning (Zhang et al., 2020a), deep latent variable models (Josse et al., 2020), and more (Li & Zhu, 2022; Yao et al., 2021a). However, these methods assume an explicit data generation process to be known, which does not hold in complex real-world scenarios.

To this end, we introduce a novel framework to mitigate unobserved confounding by generating the latent covariates using a Diffusion Probabilistic Model. We harness the exceptional generative power and training stability of diffusion models to tackle this challenging causal inference task. Specifically, we propose a conditional latent diffusion architecture that operates across different variable domains. The core of our method involves two key processes as shown in Figure 1. First, we infer a causal context vector, $u$, from the complete set of observed data $\eta^{(0)} = (X, A, Y)$. This vector encapsulates the domain-specific causal knowledge distilled from observational data, particularly pertaining to the generation of confounding variables. Second, the context vector $u$ conditions a reverse diffusion process that generates the latent covariate $Z$ by progressively denoising a vector sampled from a simple Gaussian distribution. To facilitate end-to-end training of this complex generative process, we derive a tractable Variational Lower Bound (VLB) on the log-likelihood, which provides a stable and principled optimization objective. Furthermore, to ensure the generated variable is causally valid, we introduce an Identifiability Analysis theory to guarantee that the latent variable $Z$ learned by our model is orthogonal-identifiable.

The main contributions of this paper can be concluded as follows: (1) We propose to solve the task of latent covariates in causal inference with the diffusion model; (2) To realize the above idea, we first design a novel conditional latent diffusion framework and derive a variational lower bound of the likelihood of the latent covariates conditional on the causal context vector, and then reformulate that bound into a tractable expression in closed form; (3) We theoretically analyze that the latent variable $z$ learned by our model is orthogonal-identifiable; (4) We verify the effectiveness and generality of our framework by comparing it with 12 state-of-the-art methods on two benchmarks. The empirical studies manifest that the proposed method can achieve competitive gains.

## 2 RELATED WORK

The Conditional Average Treatment Effect (CATE), also known as the Heterogeneous Treatment Effect (HTE), refers to the average treatment effects of a treatment/intervention on pre-specified outcomes for subgroups characterized by distinct covariates. Statistical methods for estimating CATE include matching Dehejia & Wahba (2002), stratification O'Muircheartaigh & Hedges (2014), reweighting Rosenbaum (1987); Bang & Robins (2005), and tree-based approaches like BART and causal forest Chipman et al. (2010); Wager & Athey (2018). Recent work introduced highly efficient deep learning algorithms to estimate CATE. The deep learning-based methods for CATE estimation can be broadly divided into two main categories: representation learning-based and generative model-based approaches. Representation learning-based methods aim to find balanced covariate representations that eliminate the differences within the distributions between treatment and control groups, thus mitigating confounding bias Assaad et al. (2021); Yao et al. (2021b); Guo et al. (2020). To achieve this, methods such as integral probability metric (IPM) regularization Johansson et al. (2016); Shalit et al. (2017), local similarity preservation Yao et al. (2018; 2019), targeted learning Shi et al. (2019); Zhang et al. (2020b), and optimal transport Wang et al. (2023); Torous et al. (2021) are employed to learn the balanced representations. On the other hand, generative model-based methods estimate counterfactual outcomes by modeling the data generation process with generative models Zhang et al. (2021); Zou et al. (2020); Guo et al. (2020). For instance, CEVAE applies variational autoencoders (VAE) to infer latent covariates from observed data Louizos et al. (2017), while SCIGAN employs generative adversarial networks (GAN) to generate missing counterfactual outcomes and combines these with factual outcomes for CATE estimation Bica et al. (2020). Different from the representation learning-based methods, the generative model-based methods allow unmeasured covariates. However, the existing generative model-based methods to estimate CATE build on strict assumptions of the data generation process, which restricts the application of these methods in real-world scenarios.

Additionally, Latent confounding arises when unmeasured variables influence both the treatment and outcome, potentially leading to biased CATE estimates Ananth & Schisterman (2018). These methods include instrumental variables, and front-door adjustment. The instrumental variable methods exploit

external instruments to account for unmeasured confounding in observational studies Imbens (2014); Wu et al. (2022), but they generally assume linear relationships and require unconfounded instruments, limiting their applicability in practice Frauen & Feuerriegel (2022). Front-door adjustment methods, on the other hand, estimate causal effects by leveraging a causal pathway (the front-door criterion) that blocks the influence of unmeasured covariates Bellemare et al. (2020); Fulcher et al. (2020). However, these methods typically require knowledge of the true causal graph, which may not always be available Shah et al. (2024); Li et al. (2024).

## 3 PRELIMINARIES

In this section, we first introduce the context of estimating CATE, understanding the underlying mechanisms of data generation and transformation, and then present the Diffusion Denoising Probabilistic Model (DDPM) framework.

### 3.1 ESTIMATION OF CONDITIONAL AVERAGE TREATMENT EFFECT

We aim to estimate the conditional average treatment effect (CATE) from the samples, which is defined as:

$$\tau(x) = \mathbb{E}[Y_1 - Y_0 \mid X = x].$$

where $Y_a$ represents the potential outcome under treatment $a$, and $x$ denotes the covariates or characteristics of the individual. This measure quantifies the expected difference in outcomes when the treatment is applied versus when it is not, conditioned on the individual's characteristics.

Let $\Phi : \mathcal{X} \times \mathcal{Z} \rightarrow \mathcal{R}$ be a representation function, $f : \mathcal{R} \times \{0, 1\} \rightarrow \mathcal{Y}$ be a hypothesis predicting the outcome of a patient's covariates $x$, given the representation covariates $\Phi(\boldsymbol{x})$ and the treatment assignment $a$. Let $L : \mathcal{Y} \times \mathcal{Y} \rightarrow \mathbb{R}_+$ be a loss function. The estimation of the potential outcome $Y(T = a) = f(\Phi(\boldsymbol{x}), a)$ ($a \in \{0, 1\}$). To identify the CATE from observed data, we require some additional assumptions. For more details about these assumptions, see the Appendix.

### 3.2 DIFFUSION MODEL

DDPMs simulate the data generation process by reversing a diffusion process that transforms real data $\boldsymbol{x}^0$ into Gaussian noise $\boldsymbol{x}^T$ over time (Ho et al., 2020). The process $p_\theta(\boldsymbol{x}^0)$ is defined as:

$$p_\theta(\boldsymbol{x}^0) = \int p(\boldsymbol{x}^T) \prod_{t=1}^{T} p_\theta(\boldsymbol{x}^{t-1} \mid \boldsymbol{x}^t) \, d\boldsymbol{x}^{1:T}$$

The sequence $\boldsymbol{x}^{T:0}$ is defined as a Markov chain with learned Gaussian transitions, each denoted by:

$$p_\theta(\boldsymbol{x}^{t-1} \mid \boldsymbol{x}^t) = \mathcal{N}(\mu_\theta(\boldsymbol{x}^t, t), \Sigma_\theta(\boldsymbol{x}^t, t)) \tag{1}$$

This formulation shows how the model uses parameterized Gaussian transitions to reverse the diffusion process step-by-step, recreating the initial data from pure noise.

**Forward Process (Diffusion).** In the forward process, starting with the data sample $\boldsymbol{x}^0$ from the distribution $q(\boldsymbol{x}^0)$, noise is incrementally added over $T$ time steps, until the data is completely converted into Gaussian noise $\boldsymbol{x}^T$. The noise addition at each step $t$ is defined by:

$$q\left(\boldsymbol{x}^{(t)} \mid \boldsymbol{x}^{(t-1)}\right) = \mathcal{N}\left(\boldsymbol{x}^{(t)}; \sqrt{\bar{\alpha}_t}\boldsymbol{x}^{(0)}, (1 - \bar{\alpha}_t)\boldsymbol{I}\right) \tag{2}$$

where $\bar{\alpha}^t = \prod_{i=1}^{t} \alpha^i$, and $\alpha^t = 1 - \beta^t$ represents how much of the previous data is retained (with $\beta^t \in (0, 1)$ is a hyper-parameter ). The $\alpha^t$ terms are crucial as they determine the rate at which the data is corrupted by noise.

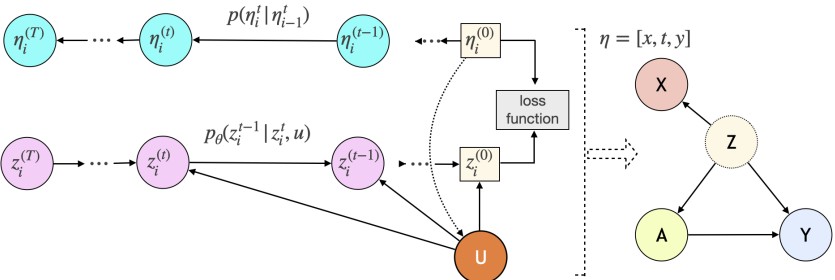

Figure 1: Corresponding graph model of our method: $x$ is the proxy variable, $z$ is latent covariate.

**Reverse Process (Denoising).** Recall that we use Equation 1 to denoise. Typically, the mean is calculated using the expression derived by the reparameterization trick and Bayes' rule:

$$\mu_\theta(\boldsymbol{x}^t, t) = \frac{1}{\sqrt{\alpha^t}} \left( \boldsymbol{x}^t - \frac{\beta^t}{\sqrt{1 - \bar{\alpha}^t}} \boldsymbol{\epsilon}_\theta(\boldsymbol{x}^t, t) \right),$$

where $\bar{\alpha}^t = \prod_{i=1}^{t} \alpha^i$. In this process, $\boldsymbol{\epsilon}_\theta(\boldsymbol{x}^t, t)$ represents the noise estimated by the parameterized network. This equation facilitates the step-by-step transformation from pure noise back to structured data. The covariance matrix is typically fixed to $\beta^t \mathbf{I}$ in practice.

## 4 DIFFUSION MODEL FOR LATENT COVARIATES

In this section, we develop a diffusion model to generate latent covariates. To perform valid latent covariates from observed datasets, we introduce a causal context vector variable u that encapsulate some essential information required to associate the observed data with its corresponding unobserved covariate.

### 4.1 INFERENCE LATENT COVARIATES USING DIFFUSION MODEL

The process of conditional image generation using diffusion models has been extensively explored (Luo & Hu, 2021; Zhang et al., 2023; Ni et al., 2023). Unlike the well-documented generation of images where generated outputs can be directly compared with training data, the generation of latent covariate $\boldsymbol{Z}$ presents unique challenges due to the absence of observable data for $\boldsymbol{Z}$. This issue necessitates the development of effective representations for unobserved variables. We address the challenge of learning these representations in Section 4.2. The formulation begins with using observed data $\boldsymbol{X}, \boldsymbol{A}, \boldsymbol{Y}$ to infer latent covariate $\boldsymbol{Z}$ through a diffusion model, i.e. modeling $p(\boldsymbol{z}|\boldsymbol{x}, \boldsymbol{a}, \boldsymbol{y})$, as shown in Figure 1.

To generate latent covariates $\boldsymbol{Z}$ requires the intrinsic connection between observed and unobserved data to infer the latent covariates from observed covariates. This motivates us to assume a hidden factor that bridges the gap. Thus, we propose introducing a causal context vector $\boldsymbol{u}$ which is learnable from the observed data to capture the domain-specific causal knowledge of the observed variable $\boldsymbol{X}$ and the unobserved variable $\boldsymbol{Z}$. For example, $\boldsymbol{u}$ can be the common knowledge between $\boldsymbol{X}$ and $\boldsymbol{Z}$. Subsequently, we can use the observed data to infer the posterior $p(\boldsymbol{u} \mid \boldsymbol{x}, \boldsymbol{a}, \boldsymbol{y})$ and then generate the corresponding $\boldsymbol{z}$ using the likelihood $p(\boldsymbol{z} \mid \boldsymbol{u}, \boldsymbol{x}, \boldsymbol{a}, \boldsymbol{y})$.

The forward diffusion process in our task involves incrementally adding noise to the observed variable $\boldsymbol{\eta}^{(0)} \sim P(X, A, Y)$, transforming the initial distribution into a pure noise distribution. This transformation occurs incrementally over $T$ steps, culminating in $\boldsymbol{\eta}^{(T)}$. This procedure adheres to the standard diffusion process outlined in Section 3.2.

In our generation process, the reverse diffusion is capable of approximating $p_{\boldsymbol{\theta}}(\boldsymbol{z}^{(t-1)}|\boldsymbol{z}^{(t)}, \boldsymbol{u})$ from a simple noise distribution $p_{\boldsymbol{\theta}}(\boldsymbol{z}^{(T)})$ that are given as the input. Therefore, with the latent variable $\boldsymbol{u}$ and the preserved information from the forward diffusion process, we can generate the desired latent

covariates $\boldsymbol{z} = z^{(0)}$ through the reverse Markov chain. Formally, the reverse diffusion process for generating latent covariates is:

$$p_{\boldsymbol{\theta}}(\boldsymbol{z}^{(0:T)}|\boldsymbol{u}) = p(\boldsymbol{z}^{(T)}) \prod_{t=1}^{T} p_{\boldsymbol{\theta}}(\boldsymbol{z}^{(t-1)}|\boldsymbol{z}^{(t)}, \boldsymbol{u}) \tag{3}$$

where $p_{\boldsymbol{\theta}}(\boldsymbol{z}^{(t-1)}|\boldsymbol{z}^{(t)}, \boldsymbol{u})$ is learnable transition kernel and $\boldsymbol{\theta}$ is the model parameters. It describes the denoising process at some time steps. The learnable transition kernel takes the form of

$$p_{\boldsymbol{\theta}}(\boldsymbol{z}^{(t-1)}|\boldsymbol{z}^{(t)}, \boldsymbol{u}) = \mathcal{N}(\boldsymbol{z}^{(t-1)}; \mu_{\boldsymbol{\theta}}(\boldsymbol{z}^{(t)}, t, \boldsymbol{u}), \beta_t \boldsymbol{I})) \tag{4}$$

In this model, the mean $\mu_{\boldsymbol{\theta}}(\boldsymbol{\eta}^{(t)}, t, \boldsymbol{u})$ are parameterized by deep neural networks learned in the optimization process and $\boldsymbol{u}$ is the latent variable encoding the shared information such as the correlation between observed and unmeasured covariates. Unlike the setup described in Section 3.2, the additional variable $\boldsymbol{u}$ establishes the dependence between $\boldsymbol{\eta}$ and $\boldsymbol{z}$, facilitating the inference of the posterior $q_{\boldsymbol{\varphi}}(\boldsymbol{u} \mid \boldsymbol{\eta}^{(0)})$ and enabling the sampling of unobserved variable $\boldsymbol{Z}$ accordingly.

In practice, we assume the noise distribution $p(\boldsymbol{\eta}^{(T)})$ to be a standard normal distribution $\mathcal{N}(0, \boldsymbol{I})$. By applying the reverse Markov chain which given the generation factors and initial distribution $p(\boldsymbol{\eta}^{(T)})$, we can retrieve the latent covariates aligned with the target distribution.

**Inference of latent covariates.** With the above well-defined denoising process established, we can now apply it to causal inference. As depicted in Algorithm 1 and Figure 1, each time we observe a data point $\boldsymbol{\eta}$, the process starts by calculating the posterior $q_{\boldsymbol{\varphi}}(\boldsymbol{u} \mid \boldsymbol{\eta}^{(0)})$, which models the latent representation $\boldsymbol{u}$ given the observed data. Subsequently, the algorithm samples a point $\boldsymbol{z}^{(T)}$ from a standard normal distribution $\mathcal{N}(0, I)$, initializing the reverse diffusion sequence. This sampled data point serves as the basis for the reverse diffusion process, which iteratively estimates $\boldsymbol{z}^{(t-1)}$ from $\boldsymbol{z}^{(t)}$ using the transition kernel $p_{\boldsymbol{\theta}}$ conditioned on $\boldsymbol{u}$. This iterative process proceeds until $t = 1$, finally yielding the inferred latent covariates $\boldsymbol{z}^{(0)}$. These covariates, alongside the initial observation $\boldsymbol{x}$, allow the model to predict the potential outcomes $y_i$ as outlined in the Figure 1. The model thus leverages both observed and latent variables to generate comprehensive predictions that integrate both observed characteristics and inferred unobserved factors.

**Variational Lower Bound.** With the formulated forward and reverse diffusion processes for latent covariates in mind, we now aim to formalize the training objective. Since directly optimizing the exact log-likelihood is intractable, we instead maximize its variational lower bound (VLB)(the detailed derivation is present in the Appendix):

$$\mathbb{E}[-\log p_{\boldsymbol{\theta}}(\boldsymbol{z}^{(0)})] \leq \underbrace{E_q \left[ \log \frac{q(\boldsymbol{\eta}^{(1:T)}, \boldsymbol{u}|\boldsymbol{\eta}^{(0)})}{p_{\boldsymbol{\theta}}(\boldsymbol{z}^{(0:T)}, \boldsymbol{u}))} \right]}_{VLB} \tag{5}$$

where $L_{VLB}$ is a common objective for training probabilistic generative models (Luo & Hu, 2021; Ho et al., 2020; Yang et al., 2023).

We can further derive the $L_{VLB}$ as:

$$L_{VLB} = E_q \left[ \sum_{t=2}^{T} D_{KL} \left( \underbrace{q(\boldsymbol{\eta}^{(t-1)}|\boldsymbol{\eta}^{(t)}, \boldsymbol{\eta}^{(0)})}_{A} \| \underbrace{p_{\boldsymbol{\theta}}(\boldsymbol{z}^{(t-1)}|\boldsymbol{z}^{(t)}, \boldsymbol{u})}_{B} \right) \right.$$
$$\left. - \log \underbrace{p_{\boldsymbol{\theta}}(\boldsymbol{z}^{(0)}|\boldsymbol{z}^{(1)}, \boldsymbol{u})}_{C} + D_{KL} \left( \underbrace{q_{\boldsymbol{\varphi}}(\boldsymbol{u}|\boldsymbol{\eta}^{(0)})}_{D} \| \underbrace{p(\boldsymbol{u})}_{E} \right) \right] \tag{6}$$

The above training objective can be optimized efficiently since each term in this objective is tractable. Among the terms, $q(\boldsymbol{\eta}^{(t-1)}|\boldsymbol{\eta}^{(t)}, \boldsymbol{\eta}^{(0)})$ is computed by a closed-form Gaussian (Luo & Hu, 2021; Ho et al., 2020), $p_{\boldsymbol{\theta}}(\boldsymbol{z}^{(t-1)}|\boldsymbol{z}^{(t)}, \boldsymbol{u})$ where $t \in \{1, 2, ..., T\}$ are trainable Gaussian distribution shown in Eq. 4. $q_{\boldsymbol{\varphi}}(\boldsymbol{u}|\boldsymbol{\eta}^{(0)})$ are learnable posterior distribution, which is the posterior of $\boldsymbol{u}$ after observe $\boldsymbol{\eta}^{(0)}$, aiming to encode the input observed covariates $\boldsymbol{\eta}^{(0)}$ into the distribution of the latent generation factor $\boldsymbol{u}$. We define it as: $q_{\boldsymbol{\varphi}}(\boldsymbol{u}|\boldsymbol{\eta}^{(0)}) = \mathcal{N}(\boldsymbol{u}; \boldsymbol{\mu}_{\boldsymbol{\varphi}}(\boldsymbol{\eta}^{(0)}), \sum_{\boldsymbol{\varphi}}(\boldsymbol{\eta}^{(0)}))$. $p(\boldsymbol{u})$ is the prior distribution

defined as isotropic Gaussian $\mathcal{N}(0, \boldsymbol{I})$, which is the most common choice for approximating the target distribution.

**Learning the Noise Model.** The training of DDPM involves learning the function $\boldsymbol{\epsilon}_\theta$ that can accurately predict the noise $\boldsymbol{\epsilon}$ added at each step based on the noisy data $\boldsymbol{\eta}^t$ and the step number $t$. The loss function used typically minimizes the mean squared error between the actual noise and the predicted noise:

$$\mathcal{L}(\theta) = \mathbb{E}_{\boldsymbol{\eta}^0, \boldsymbol{\epsilon}, t} \left[ \| \boldsymbol{\epsilon} - \boldsymbol{\epsilon}_\theta (\sqrt{\bar{\alpha}^t} \boldsymbol{\eta}^0 + \sqrt{1 - \bar{\alpha}^t} \boldsymbol{\epsilon}, t) \|^2 \right], \text{ where } \boldsymbol{\epsilon} \sim \mathcal{N}(0, I). \tag{7}$$

The loss function encourages the model to accurately infer the noise components that were added to the data, allowing the reverse process to effectively denoise the data.

## 4.2 Algorithm for estimating CATE

Following the above analysis, we propose a method called DFHTE ( Estimation of **H**eterogeneous **T**reatment **E**ffect Using **DiF**fusion Model), which takes into account the latent covariates to estimate the potential outcomes. We apply the unmeasured covariates to the observational studies data, and the loss function is shown as the following:

$$\min_{f, \boldsymbol{\Phi}} \ \mathbb{E}_{z \sim p_{\boldsymbol{\theta}}(\boldsymbol{z}^{(t-1)} | \boldsymbol{z}^{(t)}, \boldsymbol{u})} \left[ w \| y - f(\Phi(\boldsymbol{z}), a)) \|^2 \right] + \text{IPM}_G(\hat{p}_\Phi^{a=1}, \hat{p}_\Phi^{a=0}) \right]$$

$$s.t. \ \ p_{\boldsymbol{\theta}} = \arg \min_\theta \mathbb{E}_{\boldsymbol{\eta}^0 \sim p(X, A, Y), \boldsymbol{\epsilon} \sim \mathcal{N}(0, 1)} \left[ \| \boldsymbol{\epsilon} - \boldsymbol{\epsilon}_\theta (\sqrt{\bar{\alpha}^t} \boldsymbol{\eta}^0 + \sqrt{1 - \bar{\alpha}^t} \boldsymbol{\epsilon}, t) \|^2 \right] \tag{8}$$

where $w$ is used to compensate for the difference in treatment group size. It can be calculated by the proportion of treated units in the population, the latent covariate $z$ is derived by diffusion model, i.e., $\boldsymbol{z} \sim p_{\boldsymbol{\theta}}(\boldsymbol{z}^{(t-1)} | \boldsymbol{z}^{(t)}, \boldsymbol{u})$ where $t$ is the time step in reverse Markov chain and $u$ is the causal context vector, $\hat{p}_\Phi^{t=1}$ and $\hat{p}_\Phi^{t=0}$ are learned high-dimensional representations for treated and control groups respectively, $\text{IPM}_G(\cdot, \cdot)$ is the (empirical) integral probability metric w.r.t. a function family $G$. We adopt it to balance the treated and control distribution. In our framework, latent covariates is the key factor influencing the causal effect estimate, while the proxy variables, affected by latent covariates, do not directly determine the value of Y. By training the model based on latent covariates, we enhance its capacity to capture the true causal effect and simultaneously benefits alignment between the treated and control groups.

## 4.3 Model training

The training workflow of our proposed framework adheres to a two-stage procedure, as detailed below:

**Training Diffusion Model.** We first minimize the loss function of the diffusion model based on the observed datasets $P(X, A, Y)$, thereby imposing the model to learn the distribution of the latent covariates. Subsequently, we freeze the model parameters and accordingly to train our CATE model.

**Training CATE Model.** We train the CATE model in terms of $f$ and $\Phi$ based on the generated latent covariates. More contretely, The latent covariates $z_i$ is derived by diffusion model, i.e., $z_i \sim \mu_{\boldsymbol{\theta}}(c, t, u_i) + \beta_t \epsilon$, where $\epsilon, c \sim \mathcal{N}(0, I)$, $t$ is the time step in reverse Markov chain and $q_{\boldsymbol{\varphi}}(u_i | \boldsymbol{\eta_i})$ is the learned causal context vector. Here, we use a reparameterization trick to make the generation process feasible.

## 5 Identifiability Analysis

In this section, we analyze the identifiability of the proposed model. Our objective is to prove that, under certain assumptions, a well-defined correspondence exists between the latent representations learned by our model and their ground-truth counterparts.

**Theorem 1** (Identifiability of Latent Variable Z). *Under the assumptions H1-H4 detailed in Appendix B, which crucially include the existence of valid proxy variables, the latent variable $\tilde{\mathbf{z}}$ learned by our model is* **orthogonal-identifiable**. *That is, there exists an invertible affine transformation $\tilde{\mathbf{z}} = \mathbf{R}\mathbf{z} + \mathbf{b}$ between the learned $\tilde{\mathbf{z}}$ and the true $\mathbf{z}$, where $\mathbf{R}$ is an orthogonal matrix.*

*Proof Sketch.* The proof hinges on the framework of proximal causal inference, where the proxy variables $\mathbf{X}$ provide the necessary information to resolve the ambiguity of the unobserved confounder $\mathbf{z}$. This additional information, combined with constraints from the downstream CATE estimation task, ensures that the mapping from the observed data to the latent space is invertible. Furthermore, the VLB objective of the generative diffusion model forces the learned latent space to adhere to an isotropic Gaussian prior. By the Darmois-Skitovich theorem, these constraints collectively restrict the transformation between the true $\mathbf{z}$ and the learned $\tilde{\mathbf{z}}$ to the orthogonal group. $\qquad\square$

**Theorem 2** (Identifiability of CATE). *Given the identifiability of the latent confounder $\tilde{\mathbf{z}}$ established in Theorem 1, which serves as a sufficient adjustment set by blocking all back-door paths from treatment $A$ to outcome $Y$, the causal effect becomes identifiable from observational data by applying the back-door adjustment formula. Therefore, the Conditional Average Treatment Effect (CATE), CATE($\mathbf{x}$), is identifiable.*

*Proof Sketch.* Theorem 1 establishes that our model can identify a latent variable $\tilde{\mathbf{z}}$ that is a geometrically equivalent representation of the true confounder $\mathbf{z}$. As $\tilde{\mathbf{z}}$ serves as a sufficient adjustment set, it allows us to block the spurious back-door path between $A$ and $Y$. Consequently, we can apply the back-door adjustment formula to uniquely identify the true CATE. $\qquad\square$

The formal definitions, core assumptions, and detailed mathematical proofs for these theorems are provided in Appendix B.

## 6 Experiments

### 6.1 Experiment Setup

This section outlines our experimental approach for assessing the effectiveness of the proposed DFHTE model in estimating CATE across a variety of datasets. We conduct experiments using two benchmark datasets, ACIC 2016 (Dorie et al., 2019) and IHDP (Hill, 2011), which are commonly used in causal inference research. Additionally, DFHTE's performance is compared against a wide array of established causal inference models, ensuring a thorough validation of its capabilities in diverse scenarios. We adopt the commonly used metrics including Rooted Precision in Estimation of Heterogeneous Effect (PEHE) (Hill, 2011) and Mean Absolute Error (ATE) (Shalit et al., 2017) for evaluating the quality of CATE. Formally, they are defined as: $\sqrt{\epsilon_{PEHE}} = \sqrt{\frac{1}{n}\sum_{i=1}^{n}(\hat{\tau}_i - \tau_i)^2}, \epsilon_{ATE} = |\frac{1}{n}\sum_{i=1}^{n}(\hat{\tau}) - \frac{1}{n}\sum_{i=1}^{n}(\tau)|$, where $\hat{\tau}_i$ and $\tau_i$ stand for the predicted CATE and the ground truth CATE for the $i$-th instance respectively. The more details about the implementation of all adopted baselines and our methods and full experimental settings are presented in following Appendix.

### 6.2 Benchmarks

We conduct experiments based on two standard benchmark datasets, namely **ACIC 2016** Dorie et al. (2019) and **IHDP** Hill (2011). The **ACIC 2016.** was developed for the 2016 Atlantic Causal Inference Conference competition data. It comprises 4,802 units (28% treated, 72% control) and 82 covariates measuring aspects of the linked birth and infant death data (LBIDD). The dataset are generated randomly according to the data generating process setting. The **IHDP** introduced a semi-synthetic dataset for causal effect estimation. The dataset was based on the Infant Health and Development Program (IHDP), in which the covariates were generated by a randomized experiment investigating the effect of home visits by specialists on future cognitive scores. it consists of 747 units(19% treated, 81% control ) and 25 covariates measuring the children and their mothers.

For both **ACIC** and **IHDP**, we simulate proxy variables by generating a same-dimensional with original covariates. This new dataset aims to mimic the causal data generating process in terms of a latent covariates specified in advance. We generate the data below:

$$z \sim x, \;\; x_i'|z_i \sim \mathcal{N}(z_i, \sigma_1^2 z_i + \sigma_2^2(1-z_i)); \tag{9}$$

We sample the generation latent covariates $z$ from the original covariates $x$ and accordingly generate the proxy variables $x'$. We conduct experiments over randomly picked 100 realizations with 63/27/10

Table 1: Conditional average treatment effect estimation on IHDP and Jobs. We present each of the result with form mean ± standard deviation and we use bold fonts to label the best performance.

| Datasets | ACIC | | | | IHDP | | | |
|---|---|---|---|---|---|---|---|---|
| Metric | $\sqrt{\epsilon_{PEHE}}$ | | $\epsilon_{ATE}$ | | $\sqrt{\epsilon_{PEHE}}$ | | $\epsilon_{ATE}$ | |
| Task | In-sample | Out-sample | In-sample | Out-sample | In-sample | Out-sample | In-sample | Out-sample |
| RF | 5.24 ± 0.98 | 4.98 ± 1.5 | 1.69 ± 1.46 | 1.79 ± 1.52 | 6.22 ± 9.26 | 6.19 ± 9.41 | 0.35 ± 0.53 | 0.83 ± 1.87 |
| CF | 4.01 ± 1.33 | 4.01 ± 1.29 | 1.14 ± 0.67 | 1.14 ± 0.71 | 6.13 ± 9.04 | 6.23 ± 9.76 | 0.62 ± 1.27 | 0.86 ± 1.63 |
| T-learner | 4.03 ± 1.36 | 4.0 ± 1.35 | 1.11 ± 0.69 | 1.07 ± 0.69 | 7.37 ± 9.32 | 8.56 ± 10.04 | 2.84 ± 4.7 | 3.06 ± 5.05 |
| S-learner | 4.03 ± 1.36 | 4.0 ± 1.35 | 1.11 ± 0.69 | 1.07 ± 0.69 | 6.29 ± 9.36 | 6.02 ± 9.16 | 0.4 ± 0.67 | 0.63 ± 1.08 |
| CEVAE | 5.58 ± 1.57 | 5.59 ± 1.57 | 3.91 ± 1.38 | 3.95 ± 1.37 | 8.56 ± 8.86 | 8.37 ± 8.81 | 4.62 ± 2.0 | 4.8 ± 2.47 |
| BNN | 5.58 ± 1.56 | 5.59 ± 1.56 | 3.92 ± 1.36 | 3.95 ± 1.35 | 8.6 ± 8.83 | 8.41 ± 8.78 | 4.66 ± 1.96 | 4.85 ± 2.42 |
| DragonNet | 4.27 ± 1.26 | 4.32 ± 1.32 | 0.94 ± 0.78 | **0.91 ± 0.73** | 5.59 ± 6.85 | 6.12 ± 8.5 | 1.31 ± 1.86 | 1.44 ± 2.06 |
| GANITE | 4.29 ± 1.32 | 4.27 ± 1.33 | 3.29 ± 1.38 | 3.25 ± 1.39 | 6.86 ± 6.00 | 6.81 ± 5.92 | 4.48 ± 1.65 | 4.43 ± 1.53 |
| CFR$_{WASS}$ | 4.29 ± 1.24 | 4.32 ± 1.3 | 1.08 ± 0.67 | 1.03 ± 0.68 | 4.46 ± 5.33 | 6.09 ± 8.48 | 1.00 ± 1.82 | 1.21 ± 2.01 |
| CFR$_{MMD}$ | 4.24 ± 1.25 | 4.28 ± 1.34 | 0.91 ± 0.66 | 0.86 ± 0.64 | 4.30 ± 5.55 | 6.21 ± 8.46 | 0.95 ± 1.57 | 0.91 ± 1.20 |
| DeRCFR | 4.22 ± 1.26 | 4.29 ± 1.35 | 1.04 ± 0.86 | 0.97 ± 0.82 | 5.63 ± 7.37 | 6.33 ± 8.75 | 1.34 ± 2.09 | 1.53 ± 2.51 |
| ESCFR | 4.13 ± 1.24 | 4.17 ± 1.29 | 1.15 ± 0.65 | 1.09 ± 0.66 | 4.34 ± 5.3 | 6.24 ± 8.55 | 0.94 ± 1.44 | 1.00 ± 1.61 |
| DFITE | **3.97 ± 1.32** | **3.95 ± 1.32** | **1.04 ± 0.64** | 0.99 ± 0.64 | **2.25 ± 1.30** | **2.28 ± 1.33** | **0.24 ± 0.28** | **0.35 ± 0.39** |

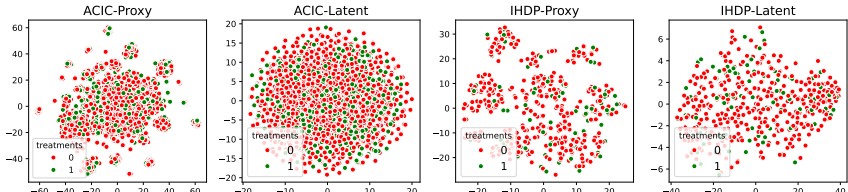

Figure 2: t-SNE visualization of the generated latent covariates z and proxy variables x condition on treatments.

train/validation/test splits by setting $\sigma_1^2$ and $\sigma_2^2$, to 0.7, 0.3 respectively.

## 6.3 OVERALL RESULTS

The overall comparison results are presented in Table 1, from which we can see: among the baselines, distance metric methods like CFR$_{WASS}$ and CFR$_{MMD}$, can obtain more performance gain both than the non-distance metric ones like GANITE and CEVAE, and traditional machine learning models like RF and CF, in most cases. This observation is consistent to our expectations and also agrees with the previous work (Shalit et al., 2017), and verify that minimizing the distance between the treated and control groups on the studied latent space can effectively eliminate the distribution shift and lead to better performance on CATE estimation.

It is encouraging to see that our model DFHTE can achieve the best performance on different datasets and evaluation metrics in more cases. The results verify the effectiveness of our idea. Comparing with the baselines, we take advantages of the latent covariates intead of proxy variables , which enable us

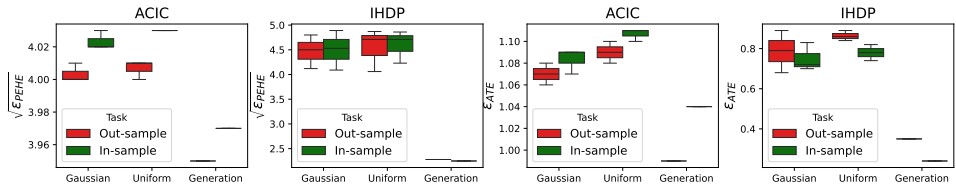

Figure 3: Performance comparison between our model and its variants on the causal context vector $\boldsymbol{u}$.

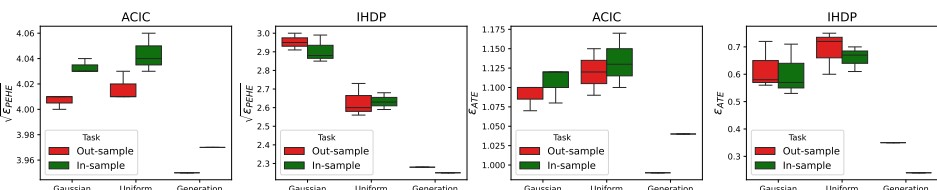

Figure 4: Performance comparison between our model and its variants on the latent covariates z.

to not only facilitate the identification of potential outcome, but also enhance to balance the studied representations between the treated and control groups. As a result, our model can always achieve the better performance on the estimation of CATE.

## 6.4 QUALITATIVE ANALYSIS

In order to provide more intuitive understandings on the generated latent covariates z, in this section, we conduct visualization studies on these latent variables, where the parameter settings follow the above experiments. From the results shown in Figure 2, we can see: The generated latent covariates Z exhibit significantly improved distributional balance compared to the original covariates X. More importantly, by conditioning on the causal context vector $u$, the generated covariates Z effectively incorporate informative priors that enhance the representational fidelity of the original feature space. These results demonstrate that structure-aware covariate generation simultaneously addresses two fundamental challenges in causal inference: (1) mitigating data sparsity through information enrichment, and (2) correcting distributional imbalances in the covariate space. Consequently, by incorporating the generated covariates into the process of CATE, this approach enables more accurate and robust treatment effect estimation.

## 6.5 COVARIATES CERTIFICATION

In this section, we would like to study whether different unobserved covariates and causal context vector in our model are necessary. To this end, we compare our model with four different unobserved covariates and causal context vectors: DFHTE(Gaussian) is a method with the unobserved covariates or causal context vector variables sampled randomly from the normal Gaussian $\mathcal{N}(0,1)$, $\mathcal{N}(1, 2.5)$, and $\mathcal{N}(-1, 2.5)$ respectively, DFHTE(Uniform) is based on Uniform $\mathcal{U}(-0.1, 0.1)$, $\mathcal{U}(-0.5, 0.5)$ and $\mathcal{U}(-1, 1)$ seperately. Both of which ara applying to the generated varible $z$ and $u$. DFHTE(Generation) is our method, in which the latent covariates $z$ are generated by a reverse diffusion model. We present the results based on $\sqrt{\epsilon_{PEHE}}$ and $\epsilon_{ATE}$ and the datasets of ACIC and IHDP. From the results shown in Figure 3 and 4, we can see: DFHTE(Gaussian) slightly performs better than DFHTE(Uniform). We speculate that the unobserved covariates sampled from normal Gaussian is more common than sampled from Uniform in practice. It is interesting to see that when we add the generated latent covariates in estimating CATE, the performance of DFHTE(Generation) is better than DFHTE(Gaussian) in all cases. This observation demonstrates the effectiveness of our idea on capturing latent covariates.

## 7 CONCLUSION

In this paper, we propose to generate the latent covariates, and accordingly to facilitate the identification of potential outcome, as well as enhancing the learned representations. To achieve this goal, we first reconstruct the latent covariates by a reverse diffusion model, and then to estimation the CATE and balance the distribution between the treated and control groups. In the experiments, we evaluate our framework based on two datasets to demonstrate its effectiveness and generality. This paper makes a first step on applying the idea of diffusion model to the field of generating latent covariates. There is still much room for improvement. To begin with, one can incorporate different prior knowledge into the generation process, and at the same time devise effective mechanism for encouraging identification to causal inference. In addition, in order to investigate the time-consuming, people can also investigate the specific time step in generating latent covariates.

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

# A  BACKGROUND: HETEROGENEOUS TREATMENT EFFECT

Under the Neyman-Rubin potential outcomes framework (Rubin, 2005), CATE estimation aims to measure the causal effect of a treatment or intervention $a \in \mathcal{A}$ on the outcome $y \in \mathcal{Y}$ for given the unit's covariates or descriptions $x \in \mathcal{X}$. Throughout this paper, we only focus on the binary treatment case, where $\mathcal{A} = \{0, 1\}$, $y$ represents the factual outcome. We treat units which received treatment, i.e., $a = 1$ as treated units and the other units with $a = 0$ as control units. The Conditional Average Treatment Effect (CATE) for unit $x$ is (Shalit et al., 2017):

$$\tau(x) := \mathbb{E}[Y_1 - Y_0 | x] \tag{10}$$

Where $Y_a$ denotes the potential outcome for treatment $a$. In practice, we can only observe the factual outcome with respect to treatment assignment, i.e., $y = Y_0$ if $a = 0$, otherwise $y = Y_1$. Usually, we build on three significant assumptions to guarantee that the potential outcomes are identifiable from observational study.

**Assumption 1. Consistency.** For a given patient with treatment assignment $a$, then the potential outcome for the treatment $a$ is the same as the observed (factual) outcome: $Y_a = y$

**Assumption 2. Positivity (Overlap) .** if $P(X = x) \neq 0$, then $P(A = a | X = x) > 0$, $\quad \forall a$ and $x$.

**Assumption 3. Strong ignorability.** For a given patient $(i)$, the treatment are independent of the potential outcomes if given the covariates $X : A \perp\!\!\!\perp Y_1, Y_0 | X$.

With these assumptions in mind, the estimation on potential outcomes could be transformed into identifiable estimation from a statistical point of view. In other words, we call that the counterfactual outcomes can be identified under these assumptions, i.e, $\tau(x) = \mathbb{E}[Y|X = x, A = 1] - \mathbb{E}[Y|X = x, A = 0]$. From machine learning perspective, these observational dataset can be modeled via a standard supervised learning model, such as SVM, for estimating $\tau(x)$. However, this model could be unreliable and unviable employed to estimate the future counterfactual outcomes under the fact that without adjusting for the bias introduced by the latent covariates and imbalanced distribution between treated groups and control groups. The existing generative-based models can achieve promising results in generating latent covariates (Louizos et al., 2017) and counterfactuals (Yoon et al., 2018), which indeed eliminate the influence from backdoor between treatment and outcome. However, they have some inherent limitations, which would hinder the model's flexibility and performance. In this paper, we build on the prominent diffusion model to generate the latent covariates, and accordingly align the distribution between treated groups and control groups and measure the CATE. We proceed in two steps: (1) Generate the latent covariates conditioned on generation factor; (2) Balance the confounder's representation in latent space and measuring the CATE based on the observed and latent covariates.

# B  DETAILED PROOF FOR IDENTIFIABILITY

## B.1  PROBLEM SETUP AND DEFINITIONS

Let $p_{data}(\boldsymbol{\eta}^{(0)})$ be the true distribution of the observed data $\boldsymbol{\eta}^{(0)} = (\mathbf{X}, A, Y)$. We assume the existence of a true, unobserved confounder $\mathbf{z} \in \mathbb{R}^{d_z}$ with a prior distribution $p^*(\mathbf{z})$. Crucially, we assume that the observed covariates $\mathbf{X}$ act as **proxy variables** for $\mathbf{z}$, as depicted in the causal graph in the main text. The true data generating process is described by a joint distribution $p^*(\mathbf{z}, \boldsymbol{\eta}^{(0)}) = p^*(\boldsymbol{\eta}^{(0)}|\mathbf{z})p^*(\mathbf{z})$.

Our model is parameterized by $\psi = (\phi, \theta)$. We abstract the entire process of inferring the "clean" latent variable $\tilde{\mathbf{z}}$ from the observed data $\boldsymbol{\eta}^{(0)}$ as a mapping $M_\psi : \mathcal{H} \to \mathcal{Z}$, where $\tilde{\mathbf{z}} = M_\psi(\boldsymbol{\eta}^{(0)})$.

**Definition 1** (Equivalent Models). Two models, parameterized by $\psi$ and $\tilde{\psi}$, are defined as equivalent (denoted $\psi \sim \tilde{\psi}$) if they both perfectly minimize the joint optimization objective. This implies that their generated marginal distributions of the observed data are identical to the true data distribution, and the marginal distribution of the generated $\tilde{\mathbf{z}}$ matches the prescribed prior.

**Definition 2** ($\mathcal{G}$-Identifiability). We say that a latent variable $\mathbf{z}$ is $\mathcal{G}$-identifiable, where $\mathcal{G}$ is a transformation group, if for any two equivalent models $\psi \sim \tilde{\psi}$, there exists a transformation $g \in \mathcal{G}$ such that the inferred latent variables $\tilde{\mathbf{z}} = M_\psi(\boldsymbol{\eta}^{(0)})$ and $\hat{\mathbf{z}} = M_{\tilde{\psi}}(\boldsymbol{\eta}^{(0)})$ satisfy $\hat{\mathbf{z}} = g(\tilde{\mathbf{z}})$ almost everywhere.

Our goal is to prove that, within our model's framework, $\mathbf{z}$ is identifiable up to a subgroup of the affine group (the orthogonal group).

## B.2 CORE ASSUMPTIONS

**H1 (Prior Form)** The true unobserved confounder $\mathbf{z}$ follows an isotropic standard normal distribution, i.e., $\mathbf{z} \sim \mathcal{N}(\mathbf{0}, \mathbf{I})$. This is a common assumption, implying that the true confounding factors are independent and identically scaled under some basis.

**H2 (Information Preservation & Proxy Validity)** The proxy variables $\mathbf{X}$ are valid in the sense of proximal causal inference. This implies that they are descendants of $\mathbf{z}$ and are separated from $Y$ by $\mathbf{z}$. This validity ensures that the true generative mapping from $\mathbf{z}$ to $\boldsymbol{\eta}^{(0)}$ is such that an invertible mapping from the observed data $\boldsymbol{\eta}^{(0)}$ back to $\mathbf{z}$ exists. The information provided by the proxies $\mathbf{X}$ is critical for resolving the ambiguity of $\mathbf{z}$ and enabling this inversion.

**H3 (Model Capacity)** All neural networks within the model belong to the class of universal function approximators, possessing sufficient capacity to fit any continuous function.

**H4 (Downstream Task Constraint)** The joint optimization objective, which combines an outcome prediction loss with an Integral Probability Metric (IPM) term for balancing the treated and control distributions, i.e., $\mathbb{E}[w\|y - f(\Phi(\mathbf{z}), a))\|^2] + \text{IPM}_G(\hat{p}_\Phi^{a=1}, \hat{p}_\Phi^{a=0})$, collectively imposes a strong constraint. To achieve the minimum loss, the generated $\mathbf{z}_i$ must contain all information originating from the true $\mathbf{z}$ that is necessary for predicting the outcome and balancing the representations.

## B.3 FORMAL PROOF OF THEOREM 1

*Proof.* The proof consists of two main steps, each established by a lemma.

**Lemma 1.** *Under assumptions H2, H3, and H4, the mapping $M_\psi : \boldsymbol{\eta}^{(0)} \mapsto \tilde{\mathbf{z}}$ learned by an optimal model must be an invertible function of the true inverse mapping $h^* : \boldsymbol{\eta}^{(0)} \mapsto \mathbf{z}$. That is, there exists an invertible function $g$ such that $\tilde{\mathbf{z}} = g(\mathbf{z})$.*

*Proof of Lemma 1.* Consider an optimal model with parameters $\psi^*$. By **H4**, the downstream task losses (outcome prediction and IPM balancing) are minimized. By **H2**, the true confounder $\mathbf{z}$ is a function of the observed data $\boldsymbol{\eta}^{(0)}$, i.e., $\mathbf{z} = h^*(\boldsymbol{\eta}^{(0)})$. The key insight from proximal causal inference is that the proxy variables $\mathbf{X} \subset \boldsymbol{\eta}^{(0)}$ provide sufficient constraints to make the inverse mapping $h^*$ unique.

We proceed by contradiction. Assume that $M_{\psi^*}$ is not an invertible function of $h^*$. This implies there exist two distinct true latent values, $\mathbf{z}_1 \neq \mathbf{z}_2$, which produce different observed data $\boldsymbol{\eta}_1^{(0)} \neq \boldsymbol{\eta}_2^{(0)}$, but are mapped to the same latent representation: $\tilde{\mathbf{z}}' = M_{\psi^*}(\boldsymbol{\eta}_1^{(0)}) = M_{\psi^*}(\boldsymbol{\eta}_2^{(0)})$.

Without the proxy variables $\mathbf{X}$, this scenario is possible and leads to non-identifiability. However, the presence of valid proxies in $\boldsymbol{\eta}^{(0)}$ means that the observational distributions for $\mathbf{z}_1$ and $\mathbf{z}_2$ will differ in a way that allows them to be distinguished. If the model maps them to the same $\tilde{\mathbf{z}}'$, it is discarding the identifying information provided by the proxies. This would lead to an inability to correctly predict the distinct potential outcomes associated with $\mathbf{z}_1$ and $\mathbf{z}_2$, preventing the downstream loss from reaching its theoretical minimum. This contradicts the assumption that the model is optimal. Therefore, to satisfy the downstream task constraint in the presence of valid proxies, the learned mapping $M_{\psi^*}(\boldsymbol{\eta}^{(0)})$ must be invertible. $\qquad\square$

**Lemma 2.** *Under assumptions H1 and H3, the marginal distribution of the latent variable $\tilde{\mathbf{z}}$ generated by an optimal model must match the prescribed prior distribution, $\mathcal{N}(\mathbf{0}, \mathbf{I})$.*

*Proof of Lemma 2.* The VLB objective forces the aggregate posterior of the learned latent variable to match the prior, regardless of the conditioning information used to generate it. The aggregate posterior is defined as:

$$\tilde{p}(\tilde{\mathbf{z}}) = \int p_{data}(\boldsymbol{\eta}^{(0)}) q_\psi(\tilde{\mathbf{z}}|\boldsymbol{\eta}^{(0)}) d\boldsymbol{\eta}^{(0)} \tag{11}$$

For the VLB to be optimal, the aggregate posterior must match the prior, i.e., $\tilde{p}(\tilde{\mathbf{z}}) = p(\tilde{\mathbf{z}}) = \mathcal{N}(\mathbf{0}, \mathbf{I})$. $\qquad\square$

**Final Proof of Theorem 1.**

Combining **Lemma 1** and **Lemma 2**, we know that the learned latent variable $\tilde{\mathbf{z}}$ is an invertible function of the true $\mathbf{z}$, and both must follow a standard normal distribution. By the **Darmois-Skitovich theorem**, the function relating them must be affine: $\tilde{\mathbf{z}} = \mathbf{R}\mathbf{z} + \mathbf{b}$, where $\mathbf{R}$ is an invertible matrix and $\mathbf{b}$ is a vector.

Finally, we determine the specific forms of $\mathbf{R}$ and $\mathbf{b}$ by matching their means and covariances. Matching the means: $\mathbb{E}[\tilde{\mathbf{z}}] = \mathbb{E}[\mathbf{R}\mathbf{z} + \mathbf{b}] = \mathbf{R}\mathbb{E}[\mathbf{z}] + \mathbf{b}$. Since $\mathbb{E}[\tilde{\mathbf{z}}] = \mathbf{0}$ and $\mathbb{E}[\mathbf{z}] = \mathbf{0}$, we have $\mathbf{b} = \mathbf{0}$. Matching the covariances: $\mathrm{Cov}(\tilde{\mathbf{z}}) = \mathrm{Cov}(\mathbf{R}\mathbf{z} + \mathbf{b}) = \mathbf{R}\mathrm{Cov}(\mathbf{z})\mathbf{R}^T$. Since $\mathrm{Cov}(\tilde{\mathbf{z}}) = \mathbf{I}$ and $\mathrm{Cov}(\mathbf{z}) = \mathbf{I}$, we have:

$$\mathbf{I} = \mathbf{R}\mathbf{I}\mathbf{R}^T = \mathbf{R}\mathbf{R}^T \tag{12}$$

The relation $\mathbf{R}\mathbf{R}^T = \mathbf{I}$ is the definition of an orthogonal matrix. In conclusion, the learned latent space $\tilde{\mathbf{z}}$ is equivalent to the true latent space $\mathbf{z}$ up to an orthogonal transformation and a displacement. $\quad\square$

### B.4 FORMAL PROOF OF THEOREM 2

*Proof.* Theorem 1 establishes that we can recover $\mathbf{z}$ up to an orthogonal transformation, i.e., $\tilde{\mathbf{z}} = \mathbf{R}\mathbf{z} + \mathbf{b}$. We now prove this ambiguity does not affect the estimation of $\mathrm{CATE}(\mathbf{x})$.

According to the back-door adjustment formula, the interventional expectation is:

$$\mathbb{E}[Y|\mathbf{X} = \mathbf{x}, \mathrm{do}(A = a)] = \int_{\mathbf{z}} \mathbb{E}[Y|\mathbf{X} = \mathbf{x}, A = a, \mathbf{z}' = \mathbf{z}]p(\mathbf{z}' = \mathbf{z}|\mathbf{X} = \mathbf{x})d\mathbf{z} \tag{13}$$

In our model, all quantities are estimated based on the learned latent variable $\tilde{\mathbf{z}}$. Thus, we compute:

$$\mathbb{E}_\psi[Y|\mathbf{X} = \mathbf{x}, \mathrm{do}(A = a)] = \int_{\tilde{\mathbf{z}}} \mathbb{E}_\psi[Y|\mathbf{X} = \mathbf{x}, A = a, \tilde{\mathbf{z}}' = \tilde{\mathbf{z}}]p_\psi(\tilde{\mathbf{z}}' = \tilde{\mathbf{z}}|\mathbf{X} = \mathbf{x})d\tilde{\mathbf{z}} \tag{14}$$

where $\mathbb{E}_\psi$ and $p_\psi$ are functions defined by the trained model. Since $\tilde{\mathbf{z}} = \mathbf{R}\mathbf{z} + \mathbf{b}$ is an invertible affine transformation, we perform a change of variables. Let $\tilde{\mathbf{z}}' = \mathbf{R}\mathbf{z}' + \mathbf{b}$, then $d\tilde{\mathbf{z}}' = |\det(\mathbf{R})|d\mathbf{z}'$. Since $\mathbf{R}$ is orthogonal, $|\det(\mathbf{R})| = 1$.

$$\mathbb{E}_\psi[Y|\mathbf{X} = \mathbf{x}, \mathrm{do}(A = a)] = \int_{\mathbf{z}} \mathbb{E}_\psi[Y|\mathbf{X} = \mathbf{x}, A = a, \tilde{\mathbf{z}}' = \mathbf{R}\mathbf{z}' + \mathbf{b}]p_\psi(\tilde{\mathbf{z}}' = \mathbf{R}\mathbf{z}' + \mathbf{b}|\mathbf{X} = \mathbf{x})d\mathbf{z}$$
$$\tag{15}$$

An optimal model will learn functions that are consistent with the true data generating process. The joint optimization ensures that the prediction function adapts to compensate for the transformation by $\mathbf{R}$ and $\mathbf{b}$:

- $\mathbb{E}_\psi[Y|\mathbf{X}, A, \tilde{\mathbf{z}}' = \mathbf{R}\mathbf{z}' + \mathbf{b}] = \mathbb{E}^*[Y|\mathbf{X}, A, \mathbf{z}' = \mathbf{z}]$

- $p_\psi(\tilde{\mathbf{z}}' = \mathbf{R}\mathbf{z}' + \mathbf{b}|\mathbf{X} = \mathbf{x}) = p^*(\mathbf{z}' = \mathbf{z}|\mathbf{X} = \mathbf{x})$

Substituting these into the integral, the learned interventional expectation equals the true one:

$$\mathbb{E}_\psi[Y|\mathbf{X} = \mathbf{x}, \mathrm{do}(A = a)] = \int_{\mathbf{z}} \mathbb{E}^*[Y|\mathbf{X} = \mathbf{x}, A = a, \mathbf{z}' = \mathbf{z}]p^*(\mathbf{z}' = \mathbf{z}|\mathbf{X} = \mathbf{x})d\mathbf{z} \tag{16}$$
$$= \mathbb{E}^*[Y|\mathbf{X} = \mathbf{x}, \mathrm{do}(A = a)] \tag{17}$$

Since we can uniquely identify the interventional expectation for any $a \in \{0, 1\}$, their difference, the $\mathrm{CATE}(\mathbf{x})$, is also uniquely identifiable. $\quad\square$

## C  EXPERIMENT DETAILS.

**Baselines.** We compare our model with the following 12 representative baselines: Random Forests (RF) (Breiman, 2001), Causal Forests (CF) (Wager & Athey, 2018), Causal Effect Variational Autoencoder (CEVAE) (Louizos et al., 2017), DragonNet (Shi et al., 2019), Meta-Learner algorithms S-Learner (Nie & Wager, 2021) and T-Learner (Künzel et al., 2019), Balancing Neural Network (BNN) (Johansson et al., 2016), Treatment-Agnostic Representation Network (TARNet) (Shalit et al., 2017), Estimation of Conditional average treatment effect using generative adversarial nets (GANITE) (Yoon et al., 2018) as well as CounterFactual Regression with the Wasserstein metric (CFR$_{WASS}$) (Shalit et al., 2017) and the squared linear MMD metric (CFR$_{MMD}$) (Shalit et al., 2017), along with a extension of CRF method Query-based Heterogeneous Treatment Effect estimation (QHTE) (Qin et al., 2021).

**Implementation details.**

We implement our methods based on QHTE (Qin et al., 2021). We adopt the commonly used metrics including Rooted Precision in Estimation of Heterogeneous Effect (PEHE) (Hill, 2011) and Mean Absolute Error (ATE) (Shalit et al., 2017) for evaluating the quality of CATE. Formally, they are defined as:

$$\sqrt{\epsilon_{PEHE}} = \sqrt{\frac{1}{n}\sum_{i=1}^{n}\left(\hat{\tau}_i - \tau_i\right)^2}, \epsilon_{ATE} = |\frac{1}{n}\sum_{i=1}^{n}(\hat{\tau}) - \frac{1}{n}\sum_{i=1}^{n}(\tau)| \tag{18}$$

where $\hat{\tau}_i$ and $\tau_i$ stand for the predicted CATE and the ground truth CATE for the $i$-th instance respectively. The more details about the implementation of all adopted baselines and our methods and full experimental settings are presented in following Appendix.

### C.1  IMPLEMENTATION AND EVALUATION OF THE DFHTE MODEL

We implement our methods based on QHTE (Qin et al., 2021). We use the same set of hyperparameters for DFHTE across four datasets. More precisely, we employ 3 similar fully-connected exponential-linear layers for the encoder $q_{\varphi}(\boldsymbol{u}|\boldsymbol{\eta}^{(0)})$, the transition kernel $p_{\boldsymbol{\theta}}(\boldsymbol{\eta}^{(t-1)}|\boldsymbol{\eta}^{(t)}, \boldsymbol{u})$, representation function $\Phi$, and the CATE prediction function $f$ respectively. The difference is that layer sizes are 128 for both $q_{\varphi}(\boldsymbol{u}|\boldsymbol{\eta}^{(0)})$ and $p_{\boldsymbol{\theta}}(\boldsymbol{\eta}^{(t-1)}|\boldsymbol{\eta}^{(t)}, \boldsymbol{u})$, 200 for $\Phi$, and 100 for $f$. we use Batch normalization (Ioffe & Szegedy, 2015) to facilitate training, and all but the output layer use ReLU (Rectified Linear Unit) (Agarap, 2018) as activation functions. In the main optimization objective, we set $\alpha$ and $\beta$ both to 1. We adopt the commonly used metrics including Rooted Precision in Estimation of Heterogeneous Effect (PEHE) (Hill, 2011) and Mean Absolute Error (ATE) (Shalit et al., 2017) for evaluating the quality of CATE. Formally, they are defined as:

$$\sqrt{\epsilon_{PEHE}} = \sqrt{\frac{1}{n}\sum_{i=1}^{n}\left(\hat{\tau}_i - \tau_i\right)^2}, \quad \epsilon_{ATE} = |\frac{1}{n}\sum_{i=1}^{n}(\hat{\tau}) - \frac{1}{n}\sum_{i=1}^{n}(\tau)| \tag{19}$$

where $\hat{\tau}_i$ and $\tau_i$ stand for the predicted CATE and the ground truth CATE for the $i$-th instance respectively.

## D  DETAILED DERIVATIONS.

The variational lower bound (VLB)is :

$$\mathbb{E}[-\log p_{\boldsymbol{\theta}}(\boldsymbol{z}^{(0)})] \leq \underbrace{E_q\left[\log\frac{q(\boldsymbol{\eta}^{(1:T)}, \boldsymbol{u}|\boldsymbol{\eta}^{(0)})}{p_{\boldsymbol{\theta}}(\boldsymbol{z}^{(0:T)}, \boldsymbol{u}))}\right]}_{VLB} \tag{20}$$

*Proof.* We present the detailed derivations of the Negative Log-Likelihood in Eq. 20.

$$-\log p_{\boldsymbol{\theta}}(\boldsymbol{z}^{(0)})$$

$$\leq \underbrace{-\log p_{\boldsymbol{\theta}}(\boldsymbol{z}^{(0)}) + D_{KL}(q(\boldsymbol{\eta}^{(1:T)}, \boldsymbol{u}|\boldsymbol{\eta}^{(0)})||p_{\boldsymbol{\theta}}(\boldsymbol{z}^{(1:T)}|\boldsymbol{z}^{(0)}, \boldsymbol{u}))}_{A}$$

$$\leq \log p_{\boldsymbol{\theta}}(\boldsymbol{z}^{(0)}) + \underbrace{E_q \left[\log \frac{q(\boldsymbol{\eta}^{(1:T)}, \boldsymbol{u}|\boldsymbol{\eta}^{(0)})}{p_{\boldsymbol{\theta}}(\boldsymbol{z}^{(1:T)}|\boldsymbol{z}^{(0)}, \boldsymbol{u}))}\right]}_{B}$$

$$\leq -\log p_{\boldsymbol{\theta}}(\boldsymbol{z}^{(0)}) + \underbrace{E_q \left[\log \frac{q(\boldsymbol{\eta}^{(1:T)}, \boldsymbol{u}|\boldsymbol{\eta}^{(0)})}{p_{\boldsymbol{\theta}}(\boldsymbol{z}^{(0:T)}, \boldsymbol{u}))}\right] + \log p_{\boldsymbol{\theta}}(\boldsymbol{z}^{(0)})}_{C}$$

$$\leq \underbrace{E_q \left[\log \frac{q(\boldsymbol{\eta}^{(1:T)}, \boldsymbol{u}|\boldsymbol{\eta}^{(0)})}{p_{\boldsymbol{\theta}}(\boldsymbol{z}^{(0:T)}, \boldsymbol{u}))}\right]}_{VLB} \tag{21}$$

$\square$

We can further derive the $L_{VLB}$ as:

$$L_{VLB} = E_q \left[\log \frac{q(\boldsymbol{\eta}^{(1:T)}, \boldsymbol{u}|\boldsymbol{\eta}^{(0)})}{p_{\boldsymbol{\theta}}(\boldsymbol{z}^{(0:T)}, \boldsymbol{u}))}\right]$$

$$= E_q \left[\sum_{t=2}^{T} D_{KL} \left(\underbrace{q(\boldsymbol{\eta}^{(t-1)}|\boldsymbol{\eta}^{(t)}, \boldsymbol{\eta}^{(0)})}_{A} || \underbrace{p_{\boldsymbol{\theta}}(\boldsymbol{z}^{(t-1)}|\boldsymbol{z}^{(t)}, \boldsymbol{u})}_{B}\right)\right.$$

$$\left. - \log \underbrace{p_{\boldsymbol{\theta}}(\boldsymbol{z}^{(0)}|\boldsymbol{z}^{(1)}, \boldsymbol{u})}_{C} + D_{KL} \left(\underbrace{q_{\boldsymbol{\varphi}}(\boldsymbol{u}|\boldsymbol{\eta}^{(0)})}_{D} || \underbrace{p(\boldsymbol{u})}_{E}\right)\right] \tag{22}$$

To make the objective clearer, we elaborate on the terms as follows:

[Term A]: $q(\boldsymbol{\eta}^{(t-1)}|\boldsymbol{\eta}^{(t)}, \boldsymbol{\eta}^{(0)})$ is computed by a closed-form Gaussian (Luo & Hu, 2021; Ho et al., 2020):

$$q(\boldsymbol{\eta}^{(t-1)}|\boldsymbol{\eta}^{(t)}, \boldsymbol{\eta}^{(0)}) = \mathcal{N}(\boldsymbol{\eta}^{(t-1)}; \boldsymbol{\mu}_t(\boldsymbol{\eta}^{(t)}, \boldsymbol{\eta}^{(0)}), \gamma_t \boldsymbol{I}) \tag{23}$$

where $\boldsymbol{\mu}_t(\boldsymbol{\eta}^{(t)}, \boldsymbol{\eta}^{(0)}) = \frac{\sqrt{\bar{a}_{t-1}}\beta_t}{1-\bar{a}_t}\boldsymbol{\eta}^{(0)} + \frac{\sqrt{a_t}(1-\bar{a}_{t-1})}{1-\bar{a}_t}\boldsymbol{\eta}^{(t)}$ and $\gamma_t = \frac{1-\bar{a}_{t-1}}{1-\bar{a}_t}\beta_t$.

[Terms B, C]: $p_{\boldsymbol{\theta}}(\boldsymbol{z}^{(t-1)}|\boldsymbol{z}^{(t)}, \boldsymbol{u})$ where $t \in \{1, 2, ..., T\}$ are trainable Gaussian distribution shown in Eq. 4.

[Term D]: $q_{\boldsymbol{\varphi}}(\boldsymbol{u}|\boldsymbol{\eta}^{(0)})$ are learnable posterior distribution, which is the posterior of $\boldsymbol{u}$ after observe $\boldsymbol{\eta}^{(0)}$, aiming to encode the input observed covariates $\boldsymbol{\eta}^{(0)}$ into the distribution of the latent generation factor $\boldsymbol{u}$. We define it as: $q_{\boldsymbol{\varphi}}(\boldsymbol{u}|\boldsymbol{\eta}^{(0)}) = \mathcal{N}(\boldsymbol{u}; \boldsymbol{\mu}_{\boldsymbol{\varphi}}(\boldsymbol{\eta}^{(0)}), \sum_{\boldsymbol{\varphi}}(\boldsymbol{\eta}^{(0)}))$.

[Term E]: $p(\boldsymbol{u})$ is the prior distribution defined as isotropic Gaussian $\mathcal{N}(0, \boldsymbol{I})$, which is the most common choice for approximating the target distribution.

---

**Algorithm 1:** Inference of latent covariates

1  **Input:** Observed data point $\boldsymbol{x}$.
2  Calculate the posterior $q_{\boldsymbol{\varphi}}(\boldsymbol{\eta} \mid \boldsymbol{x})$;
3  Sample data points $\boldsymbol{z}^{(T)} \sim \mathcal{N}(0, I)$;
4  Use the learned reverse process to estimate $p_{\boldsymbol{\theta}}(\boldsymbol{z}^{(t-1)} \mid \boldsymbol{z}^{(t)}, \boldsymbol{\eta})$ , $t = T, T-1, \ldots, 1$;
5  **Return:** The latent covariates $\boldsymbol{z}^{(0)}$.

---

**Algorithm 2:** Training

1  Indicate the observational data $\mathcal{X}$.
2  Initialize all the model parameters.
3  **while** *not converged* **do**
4     Sample $\boldsymbol{\eta}^{(0)} \sim \mathcal{X}$
5     Sample $\boldsymbol{\eta} \sim q_{\boldsymbol{\varphi}}(\boldsymbol{\eta}|\boldsymbol{\eta}^{(0)})$
6     Sample $t \sim \text{Uniform}(\{1, ..., T\})$
7     Sample $\boldsymbol{\eta}_1^{(t)}, ..., \boldsymbol{\eta}_m^{(t)} \sim q(x^{(t)}|x^{(0)})$
8

$$L_\theta = \sum_{i=1}^m D_{KL}\left(q(\boldsymbol{x}_i^{(t-1)}|\boldsymbol{\eta}^{(t)}, \boldsymbol{x}_i^{(0)})||p_{\boldsymbol{\theta}}(\boldsymbol{z}^{(t-1)}|\boldsymbol{z}_i^{(t)}, \boldsymbol{\eta})\right)$$

$$L_\varphi = D_{KL}\left(q_{\boldsymbol{\varphi}}(\boldsymbol{\eta}|\boldsymbol{\eta}^{(0)})||p(\boldsymbol{\eta})\right)$$

9     Compute the gradients of the $L_\theta + \frac{1}{T} L_\varphi$ Perform the gradient descent.
10  **end**

---

*Proof.* We present the detailed derivations of the VLB in Eq. 22.

$$L_{VLB} = E_q\left[\log \frac{q(\boldsymbol{\eta}^{(1:T)}, \boldsymbol{u}|\boldsymbol{\eta}^{(0)})}{p_{\boldsymbol{\theta}}(\boldsymbol{z}^{(0:T)}, \boldsymbol{\eta}))}\right]$$

$$= E_q\left[\log \frac{q(\boldsymbol{\eta}|\boldsymbol{\eta}^{(0)})\prod_{t=1}^T q(\boldsymbol{\eta}^{(t)}|\boldsymbol{\eta}^{(t-1)})}{p_{\boldsymbol{\theta}}(\boldsymbol{\eta})p(\boldsymbol{z}^{(T)})\prod_{t=1}^T p_{\boldsymbol{\theta}}(\boldsymbol{z}^{(t-1)}|\boldsymbol{z}^{(t)}, \boldsymbol{\eta})}\right]$$

$$= E_q\left[-\log p(\boldsymbol{z}^{(T)}) + \sum_{t=1}^T \log \frac{q(\boldsymbol{\eta}^{(t)}|\boldsymbol{\eta}^{(t-1)})}{p_{\boldsymbol{\theta}}(\boldsymbol{z}^{(t-1)}|\boldsymbol{z}^{(t)}, \boldsymbol{\eta})} + \log \frac{q_{\boldsymbol{\varphi}}(\boldsymbol{\eta}|\boldsymbol{\eta}^{(0)})}{p_{\boldsymbol{\theta}}(\boldsymbol{\eta})}\right]$$

$$= E_q\left[-\log p(\boldsymbol{z}^{(T)}) + \log \frac{q(\boldsymbol{\eta}^{(1)}|\boldsymbol{\eta}^{(0)})}{p_{\boldsymbol{\theta}}(\boldsymbol{z}^{(0)}|\boldsymbol{z}^{(1)}), \boldsymbol{\eta})} + \sum_{t=2}^T \log \left(\frac{q(\boldsymbol{\eta}^{(t-1)}|\boldsymbol{\eta}^{(t)}, \boldsymbol{\eta}^{(0)})}{p_{\boldsymbol{\theta}}(\boldsymbol{z}^{(t-1)}|\boldsymbol{z}^{(t)}, \boldsymbol{\eta})} \cdot \frac{q(\boldsymbol{\eta}^{(t)}|\boldsymbol{\eta}^{(0)})}{q(\boldsymbol{\eta}^{(t-1)}|\boldsymbol{\eta}^{(0)})}\right) + \log \frac{q_{\boldsymbol{\varphi}}(\boldsymbol{\eta}|\boldsymbol{\eta}^{(0)})}{p_{\boldsymbol{\theta}}(\boldsymbol{\eta})}\right]$$

$$= E_q\left[-\log p(\boldsymbol{z}^{(T)}) + \log \frac{q(\boldsymbol{\eta}^{(1)}|\boldsymbol{\eta}^{(0)})}{p_{\boldsymbol{\theta}}(\boldsymbol{z}^{(0)}|\boldsymbol{z}^{(1)}), \boldsymbol{\eta})} + \sum_{t=2}^T \log \frac{q(\boldsymbol{\eta}^{(t-1)}|\boldsymbol{\eta}^{(t)}, \boldsymbol{\eta}^{(0)})}{p_{\boldsymbol{\theta}}(\boldsymbol{z}^{(t-1)}|\boldsymbol{z}^{(t)}, \boldsymbol{\eta})} + \log \frac{q(\boldsymbol{\eta}^{(T)}|\boldsymbol{\eta}^{(0)})}{q(\boldsymbol{\eta}^{(1)}|\boldsymbol{\eta}^{(0)})} + \log \frac{q_{\boldsymbol{\varphi}}(\boldsymbol{\eta}|\boldsymbol{\eta}^{(0)})}{p_{\boldsymbol{\theta}}(\boldsymbol{\eta})}\right]$$

$$= E_q\left[-\log \frac{p(\boldsymbol{\eta}^{(T)})}{q(\boldsymbol{\eta}^{(T)}|\boldsymbol{\eta}^{(0)})} - \log p_{\boldsymbol{\theta}}(\boldsymbol{z}^{(0)}|z^{(1)}), \boldsymbol{\eta}) + \sum_{t=2}^T \log \frac{q(\boldsymbol{\eta}^{(t-1)}|\boldsymbol{\eta}^{(t)}, \boldsymbol{\eta}^{(0)})}{p_{\boldsymbol{\theta}}(\boldsymbol{z}^{(t-1)}|\boldsymbol{z}^{(t)}, \boldsymbol{\eta})} + \log \frac{q_{\boldsymbol{\varphi}}(\boldsymbol{\eta}|\boldsymbol{\eta}^{(0)})}{p_{\boldsymbol{\theta}}(\boldsymbol{\eta})}\right]$$

$$= E_q\left[\sum_{t=2}^T D_{KL}\left(q(\boldsymbol{\eta}^{(t-1)}|\boldsymbol{\eta}^{(t)}, \boldsymbol{\eta}^{(0)})||p_{\boldsymbol{\theta}}(\boldsymbol{z}^{(t-1)}|\boldsymbol{z}^{(t)}, \boldsymbol{\eta})\right) - \log p_{\boldsymbol{\theta}}(\boldsymbol{z}^{(0)}|\boldsymbol{z}^{(1)}, \boldsymbol{\eta}) + D_{KL}\left(q_{\boldsymbol{\varphi}}(\boldsymbol{\eta}|\boldsymbol{\eta}^{(0)})||p_{\boldsymbol{\theta}}(\boldsymbol{\eta})\right)\right]$$

$$\tag{24}$$

$$\square$$

# E   PSEUDO-CODE OF DFHTE

We present the diffusion model training algorithm in Algorithm 2, the sampling algorithm in Algorithm 3, and our CATE estimation algorithm in Algorithm 4.

---

**Algorithm 3:** Sampling

---

1   Sampling data points: $\boldsymbol{z}^{(T)} \sim \mathcal{N}(0, \boldsymbol{I})$.

2   **for** $t = T, ..., 1$ **do**

3        $\epsilon \sim \mathcal{N}(0, \boldsymbol{I})$ if $t > 0$, else $\epsilon = 0$

4        $\boldsymbol{z}^{(t-1)} = \mu_{\boldsymbol{\theta}}(\boldsymbol{z}^{(t)}, t, \eta) + \beta_t \epsilon$

5   **end**

6   return latent covariates $\boldsymbol{z}^{(0)}$

---

---

**Algorithm 4:** Learning algorithm of our model

---

1   Generating the latent covariates $z_1, ..., z_m$ through Algorithm 3.

2   Indicate the observational data $(x_1, z_1, t_1, y_1), ..., (x_m, z_m, t_m, y_m)$.

3   Indicate the scaling parameter $\alpha$ and $\beta$ .

4   Initialize all the model parameters.

5   Indicate the epoch number $E$.

6   Compute $u = \frac{1}{m} \sum_{i=1}^{m} t_i$.

7   Compute $w_i = \frac{t_i}{2u} + \frac{1-t_i}{2(1-u)}$ for $i = 1, ..., m$

8   **for** $e = 0$ **to** $E$ **do**

9        Sample mini-batch data $\mathcal{B}$ from $D$

10       Compute the gradients of the empirical loss:

$$g_1 = \nabla_W \frac{1}{|\mathcal{B}|} \sum_{i=1}^{|\mathcal{B}|} w_i L(y_i, f(\Phi(x_i, z_i), t_i))$$

11       Compute the gradients of the regularization:

$$g_2 = \nabla_W \beta \mathcal{R}(f)$$

12       Compute the gradients of the IPM term:

$$g_3 = \nabla_W \alpha IPM_G(\hat{p}_{\Phi}^{t=1}, \hat{p}_{\Phi}^{t=0})$$

13       Obtain the step size scalar $\rho$ with the Adam

14       Update the parameters:

$$W \leftarrow W - \rho(g_1 + g_2 + g_3)$$

15   **end**

---

Table 2: Statistics of the datasets used in our experiments.

| Dataset | #Replications | #Units | #covariates | Treated Ratio | Control Ratio |
|---------|---------------|--------|-------------|---------------|---------------|
| ACIC | 100 | 4,802 | 82 | 28% | 72% |
| IHDP | 1,000 | 747 | 25 | 19% | 81% |

