# OpenReview forum: "Mitigating Unobserved Confounding via Diffusion Probabilistic Models"
_ICLR.cc/2026/Conference — Submitted to ICLR 2026_

### Official Review · Reviewer_Xs2V · 2025-10-18

**Soundness:** 2
**Presentation:** 2
**Contribution:** 1
**Rating:** 4
**Confidence:** 5

**Summary:**

This paper introduces a novel approach for estimating the CATE from observational data, specifically tackling the critical challenge of unobserved confounders. The core innovation is the use of a diffusion model to learn the distribution of these latent confounders. By training a reverse diffusion process conditioned on observed variables, the model can effectively generate the missing confounder information. This reconstruction allows for CATE estimation in settings where the standard unconfoundedness assumption is violated. The authors support their method by deriving a tractable variational lower bound for the confounder's likelihood. Empirical results on both synthetic and real-world datasets confirm that the proposed method consistently outperforms some baselines.

**Strengths:**

The paper is clearly written.

**Weaknesses:**

1.  **Potentially Misleading Title**: The title's claim of "removing the unmeasured confounding" might overstate the paper's contribution. Given that the SUTVA assumption is made, which helps circumvent significant identifiability challenges, a title that more precisely reflects the specific problem setting and assumptions would be more appropriate.

2.  **Significant Novelty Concerns and Missing Comparison**: The novelty of this work is a major concern, especially when compared to **DiffPO (NeurIPS 2025)**. DiffPO addresses the exact same problem setup using a similar diffusion-based approach and provides strong theoretical guarantees, including Neyman orthogonality. The authors must **cite DiffPO** and provide a detailed discussion of the technical and theoretical differences to clearly delineate their own contribution.

3.  **Insufficient Empirical Evaluation**: The experimental validation is not fully convincing. The chosen **baselines are largely outdated** (e.g., TARNet variants), and more recent, relevant methods like CMGP, NSGP, and DiffPO are absent. Furthermore, the **performance gains reported are marginal**, which calls into question the practical utility of the proposed method. A runtime comparison is also missing, which is crucial for assessing the method's efficiency.

4.  **Lack of Methodological Clarity**: Several key components of the proposed method are not clearly motivated. For instance, the necessity and role of the **context vector `u`**—in addition to the latent variable `z`—are not well-explained. Similarly, the objective function in Section 4.1, which involves maximizing the log-likelihood of a **latent variable**, is counter-intuitive and requires a much clearer justification.

**Questions:**

- In line 131, the function $\Phi$ is defined over $\mathcal{X} \times \mathcal{Z}$, but the latent space $\mathcal{Z}$ has not been introduced at this point. This makes the definition unclear. In line 134, I would suggest using the subscript notation for potential outcomes, i.e., $Y_{a}$, instead of $Y(T=a)$, for clarity and consistency with much of the causal inference literature. Furthermore, to avoid ambiguity with the time step $t$ used extensively in the diffusion process, it would be beneficial to denote the treatment variable with $A$ instead of $T$ throughout the manuscript.

- In line 205, the introduction of the context vector $u$ requires further clarification. What is its specific role and motivation? Does $u$ correspond to a meaningful, recoverable variable in the data-generating process, or is it primarily an auxiliary variable for optimization? Given that the model already includes a latent vector $Z$, the necessity and advantages of introducing an additional context vector $u$ are not immediately clear. The paper would be strengthened by a more detailed explanation of why both $Z$ and $u$ are needed and what distinct roles they play. Additionally, since $u$ is introduced during the learning process and not mentioned in the DGP, clarification on its conceptual standing would be helpful.

- In Section 4.1, the objective is stated as maximizing the log-likelihood of $p(z^{(0)})$. Since $z^{(0)}$ is a latent variable, it is unclear how this is achieved directly. Could the authors elaborate on the specifics of this optimization? Furthermore, the objective in Equation 6 appears to enforce an alignment between the posterior distribution of $z$ and the distribution of $\eta$. The intuition behind this alignment and its role in the overall model is not clear. A more detailed explanation would greatly improve the reader's understanding.

- In Section 4.2, please clarify the role of $\omega$. Is it a hyperparameter serving as a weighting term? Regarding Equation 8, the procedure for calculating the loss needs more detail. Specifically, is a new $z$ sampled from the diffusion process at each training step to compute this loss? Clarifying this sampling strategy is important for understanding the computational complexity and implementation of the method.

- In Theorem 1, the terminology "orthogonal-identifiable" is non-standard. Could the authors clarify its meaning or relate it to existing concepts in the literature? Additionally, the proof for the identifiability of the CATE under the stated assumptions (e.g., unconfoundedness, overlap) seems to re-establish a well-known result in the causal inference literature. While correctness is not in question, the novelty of this specific proof is unclear. The authors might consider citing standard results and focusing the theoretical contribution on aspects more unique to their proposed model.

- In Equation 9, the notation "$z$ from $x$" is ambiguous. Could you please clarify how $z$ is obtained from $x$? Is it sampled from a posterior distribution $q(z|x)$, or is it the output of an encoder network, i.e., $z = f(x)$?

- The experimental evaluation could be significantly strengthened. The selected baseline methods appear outdated and are mostly variants of TARNet. I strongly recommend including more recent and competitive methods such as CMGP, NSGP, and especially the highly relevant DiffPO, to provide a more comprehensive and convincing comparison. Furthermore, the reported performance gains over the existing, older baselines are quite marginal, which raises questions about the practical significance of the proposed method. To better justify its contribution, could the authors highlight other advantages of DFITE? For instance, a comparison of the running times against all baselines would be very informative. Finally, please ensure the model's name is consistent throughout the paper (it appears as both DFITE and DFHTE).

---

### Official Review · Reviewer_A9kL · 2025-10-27

**Soundness:** 1
**Presentation:** 1
**Contribution:** 2
**Rating:** 2
**Confidence:** 3

**Summary:**

This paper proposes a framework using a conditional diffusion model to improve CATE estimation in a setting where the true latent confounders are unobserved but proxies that are sufficient for identification are available. The diffusion model is sued to synthesize the unobserved confounders while using an IPM term to adjust for potential distribution shift between treated and untreated populations. Further, the authors provide experiments on adjusted (semi-)synthetic datasets and compare their method to baselines from CATE estimation.

**Strengths:**

- The topic of the paper, estimating the CATE from observational data, is important for many real-world problems around decision making and thinking about unobserved confounding in terms of latent variable models is an interesting research direction.
- Code is provided for reproducibility (I did not run the code).

**Weaknesses:**

Overall, the motivation and structure of the paper is not really clear and not consistent.
- The comparison with previous work can be improved clearly. In the introduction, the related works are mixed and the narrative is not easy to follow. E.g. for motivating the necessity for methods adjusting for unobserved confounding, introducing methods for balanced representations under ignorability seems arbitrary, and the comparison to generative based methods like CEVAE, GANITE, SCIGAN does not seem accurate and is vague (e.g. l.103: “Different from the representation learning-based methods, the generative model-based methods allow unmeasured covariates” --> does not seem to be true and also the implications of this statement are unclear because then they could be used in the unobserved confounding setting?). Similar vague statements can be found throughout the paper.
- In general, the distinction between confounding bias due to unobserved confounding and finite-sample bias due to covariate shift in fully observed confounding is not elaborated clearly and the authors do not really state which challenge they actually want to tackle (the assumptions in the Appendix and the IPM term in their method imply more the latter, however, the motivation sounds more like the former which is really misleading).
- Important related work around latent variables and unobserved confounding like [1] (and related discussions about guaranteeing identifiability in such a setting), and around proxy variables, are missing.
- The Problem setting (Sec. 3.) is confusing and incomplete and the notation is inconsistent (e.g. a and t are mixed, see also Figure 1). Figure 1 implies just a proxy setting? However, this setting is barely described in the main paper but only in the Appendix, making it really hard to follow the idea of the paper.
- Identification of the true confounders given proxies is usually highly unrealistic; also I could not follow the structure of the proofs trying to ensure identifiability.
- The contribution is really limited. If I understand the paper correctly, the paper actually assumes identifiability and the major technical contribution is adding a balancing term to the training process of the diffusion model for learning the conditional distribution of potential outcomes.
- The experimental setup is unclear, especially how the proxies influence the outcome and treatment assignment (Eq. 9); the results in Fig. 3 and 4 are unclear and not explained properly.
- Minor: The citation style (direct vs indirect) needs to corrected.

[1] Wang, Yixin, and David M. Blei. "The blessings of multiple causes." Journal of the American Statistical Association 114.528 (2019): 1574-1596.

**Questions:**

See the unclarities described in the Weakness section.

---

### Official Review · Reviewer_MooC · 2025-11-01

**Soundness:** 2
**Presentation:** 1
**Contribution:** 2
**Rating:** 2
**Confidence:** 3

**Summary:**

The paper introduces a diffusion-structured model to learn from proxy variables for the classical problem of CATE.  SOTA results are reported on two datasets compared against many typical CATE methods.

**Strengths:**

- diffusion has good potential to give accurate generative models
- good results are achieved compared against many baseline methods

**Weaknesses:**

- the introduction of the new diffusion structure and causal context vector are not well-motivated in the writing
- the introduction states the other GAN and deep learning approaches assume an explicit data generation process, which is not true
- figure 1 does not clearly demonstrate the method
- figure 2 seems unnecessary
- table 1 has several typos
- a discussion on why the diffusion-based approach is able to get improved performance over previous results is not included


Formatting errors and writing issues, for example the incorrect citations, detract from the readability.

**Questions:**

- Are the ACIC and IHDP results reported on the noisy or unnoised versions of the dataset?
- Can you discuss more the necessity of the assumptions H1-H4 in proving the identifiability theorem and how realistic they are to hold on real-world data?
- What is the major advantage of the diffusion approach over existing CATE approaches?
- What is the major advantage of your diffusion approach over naive application of diffusion to CATE?

---

### Official Review · Reviewer_YR2m · 2025-11-03

**Soundness:** 1
**Presentation:** 1
**Contribution:** 1
**Rating:** 0
**Confidence:** 3

**Summary:**

The paper introduces a novel method to mitigate the unobserved confounding for conditional average treatment effect (CATE) estimation. The authors assume that the unobserved confounder is orthogonal-identifiable and use a conditional diffusion model to infer a corresponding latent variable. In a series of several semi-synthetic experiments, the effectiveness of the model was demonstrated in comparison with several other baselines.

**Strengths:**

The application of diffusion models to mitigate the unobserved confounding is an original idea (I haven’t seen it in other works so far).

**Weaknesses:**

The main flaw of the paper, in my opinion, is the lack of clear separation between identifiability and estimability assumptions. Specifically, the assumptions H1-H4 combine both the assumptions on the data-generating mechanism (= identifiability) and the assumptions on the chosen estimation model (= estimability). I encourage the authors to fully revisit the paper so that those two parts are clearly separated.  Also, the assumptions are important enough to be in the main part of the paper.

Furthermore, I had a hard time understanding what specific setting the paper operates in. In Sec. 4-5, the authors mention proxy variables, but no related work is provided about the proximal causal inference, and no relevant baselines are selected for benchmarking (e.g., [1]). On the other hand, in Appendix A, the authors made the standard ignorability assumption (=no unmeasured confounding). Also, standard CATE baselines are used in the experiments, which are originally not tailored for the proximal (or any other hidden confounding) setting.

Also, I found some smaller mistakes:

- Wrong citing style in lines 84-104 (should be \citep instead of \citet).

- Line 106. “These methods” refers to what?

- Lines 46-53. “To address the issue of unobserved confounding, some methods that only rely on large-scale observation data (OBS) … For instances, VAE-based method CEVAE …  GANITE (Yoon et al., 2018) aims to generate the counterfactual“. This is false; All the mentioned methods do not assume hidden confounding.

- Definition 1. “This implies that … ” The implications should not be a part of the definition.


References:
- [1] Sverdrup, Erik, and Yifan Cui. "Proximal causal learning of conditional average treatment effects." International Conference on Machine Learning. PMLR, 2023.

**Questions:**

- Why is balancing with integral probability metrics (IPMs) used for the main method, if it can potentially cause a representation-induced confounding bias [1]?
- Line 11. “Learning Conditional average treatment effect estimation from observational data is a challenging task due to the existence of latent covariates.” Is there a reference or a justification for such a statement? Do the authors mean unobserved confounders here? To my understanding, latent covariates != unobserved confounders in general.
- Line 14. “…overlooking the impact of an apriori knowledge on the generation process of the latent variable”. In what real-world setting can we a priori know the generation process of the latent variable?
- Line 672. What are “future counterfactual outcomes”?
- lines 671 - 674. “However, this model could be unreliable and unviable employed to estimate the future counterfactual outcomes under the fact that without adjusting for the bias introduced by the latent covariates and imbalanced distribution between treated groups and control groups.” Is there a reference/justification that we need to “adjust” for latent variables? For example, DR-learner was shown to be min-max optimal for CATE estimation (e.g., see [2]), and it does not mention any need to explicitly model latent variables.
- Lines 692-693. What are ”clean” latent variables?

References:
- [1] Melnychuk, Valentyn, Dennis Frauen, and Stefan Feuerriegel. "Bounds on representation-induced confounding bias for treatment effect estimation." arXiv preprint arXiv:2311.11321 (2023).
- [2] Jin, Jikai, Lester Mackey, and Vasilis Syrgkanis. "It's Hard to Be Normal: The Impact of Noise on Structure-agnostic Estimation." arXiv preprint arXiv:2507.02275 (2025).

---

### Meta-Review · Area_Chair_kmxZ · 2026-01-05

**Summary:**

The paper aims to tackle the conditional average treatment effect (CATE) via diffusion-based approach. Despite the interesting ideas, the paper may need further improvement on the structure and clarity. The questions raised by the reviewers were not addressed. As reviewers reach the consensus, the manuscript need further clarity improvement and structural change before publication.

**Reviewer Concerns:**

The causal context and the novel approach on diffusion-structured need to be improved;

The GAN and deep learning approach need to be further clarified;

Clarity in referring to concepts, as raised by reviewers in details; Defined concepts need to be clear;

Comparison of previous work need to be improved; missing important literature review; citation no used properly;

Several typos and formatting errors to be addressed.

**Reviewer Scores:**

The concerns were not responded so the reviewer score may not change even with full discussion.

---

### Decision · Program_Chairs · 2026-01-26

Reject